# Multi-protein assemblies orchestrate co-translational enzymatic processing on the human ribosome

Marius Klein  [1], Klemens Wild  [1] & Irmgard Sinning  [1] ✉

Nascent chains undergo co-translational enzymatic processing as soon as their N-terminus becomes accessible at the ribosomal polypeptide tunnel exit (PTE). In eukaryotes, N-terminal methionine excision (NME) by Methionine Amino-peptidases (MAP1 and MAP2), and N-terminal acetylation (NTA) by N-Acetyl-Transferase A (NatA), is the most common combination of subsequent modifications carried out on the 80S ribosome. How these enzymatic processes are coordinated in the context of a rapidly translating ribosome has remained elusive. Here, we report two cryo-EM structures of multi-enzyme complexes assembled on vacant human 80S ribosomes, indicating two routes for NME-NTA. Both assemblies form on the 80S independent of nascent chain sub-strates. Irrespective of the route, NatA occupies a non-intrusive 'distal' binding site on the ribosome which does not interfere with MAP1 or MAP2 binding nor with most other ribosome-associated factors (RAFs). NatA can partake in a coordinated, dynamic assembly with MAP1 through the hydra-like chaperon-ing function of the abundant Nascent Polypeptide-Associated Complex (NAC). In contrast to MAP1, MAP2 completely covers the PTE and is thus incompatible with NAC and MAP1 recruitment. Together, our data provide the structural framework for the coordinated orchestration of NME and NTA in protein biogenesis.

At early states of protein biogenesis, the N-terminus of the polypeptide nascent chain (NC) experiences vivid competition between different ribosome-associated factors (RAFs) which stand in line to cleave, acetylate or myristoylate these residues[1,2]. When the N-terminus starts to protrude from the polypeptide tunnel exit (PTE), the initiator methionine is in a highly predictable position while the nascent chain remains confined within the exit tunnel[3]. This unique opportunity is most prominently exploited by methionine aminopeptidases and acetylases, which must carry out N-terminal methionine excision (NME) and N-terminal acetylation (NTA) in a coordinated manner. In eukaryotes, the vast majority of proteins undergo co-translational NME, followed by NTA[1]. Both modifications are crucial, as they impinge on protein turnover, function, localization, and stability[4]. Misfunction in either machinery has been implicated in disease[5–9].

Strikingly, methionine aminopeptidases (MAPs) and N-terminal Acetyltransferases (NATs) are low in abundance relative to the number of ribosomes inside the cell[10]. Their underrepresentation necessitates well-coordinated regulation over their timely recruitment, function, and eventual displacement. The high demand for co-translational NME is attended to by two largely conserved enzymes in eukaryotes. The two eukaryotic MAP isoforms MAP1 and MAP2 share the characteristic pita-bread fold with their active dinuclear centers[11]. However, specific sequence insertions in MAP2 (including a prominent ~60 AS insert domain), as well as fundamental differences in the architecture of the N-terminal extension, elicit changes in how these factors are recruited to and accommodated on the ribosomal surface[12,13]. The zinc-finger motif containing N-terminus of mammalian MAP1 is recognized by the nascent polypeptide-associated complex (NAC), which administers

[1]Heidelberg University Biochemistry Center (BZH), Im Neuenheimer Feld 328, 69120 Heidelberg, Germany. ✉e-mail: irmi.sinning@bzh.uni-heidelberg.de

MAP1 recruitment to the PTE[13]. In contrast, our recent structures of MAP2 reveal a different, factor-independent mode of binding to the ribosome, and a more central placement on the PTE, as well as the presence of a second binding site in the ribosomal A-site[12]. Beyond its role as a co-translational protease, MAP2 has been shown to influence translation initiation[14] and might also act on other stages of the translation cycle via its A-site interaction.

About 85% of all human proteins are N-acetylated by five conserved N-acetyltransferases (NATs) (NatA-E) with distinct substrate specificities[4]. The majority is carried out by the heterodimeric NatA[4,15], which possesses the broadest substrate specificity (Gly-, Ala-, Ser-, Thr-, Val-, Cys-). NatA comprises the large scaffold protein Naa15, and the small Naa10 enzyme with the characteristic GNAT (GCN5-related-N-acetyltransferase) fold, characterized by a conserved α/β-topology[16–18]. Herein, Naa15 composes of 13 tetratricopeptide repeat (TPR) motifs (45 α-helices) which form an elaborate ring-like scaffold that engulfs the acetyltransferase[19,20]. In Saccharomyces cerevisiae (Sc), the NatA complex is exclusively present as a NatE heterotrimer with a catalytically inactive Naa50 (refs. 21,22). Unlike NatB or NatC, NatA cannot acetylate the initiator methionine, thus necessitating prior NME activity by MAP1 or MAP2 (ref. 4). As these processes must coincide within a narrow window of opportunity while translation is rapidly progressing, a functional synchronization between NME and NTA appears likely. Previously, the NatE-80S structure has been determined in yeast[23] with Naa50 strongly contributing to the 80S interaction. Therefore, the absence of this subunit might affect how human NatA engages with the ribosome.

In addition to Naa50, HypK has been identified as an important NatA interactor[24], which binds to the α-helical Naa15 scaffold. HypK composes of a three-helix bundle ubiquitin associated (UBA) domain, a long central helix, and a mostly disordered extended N-terminus. In complex with NatA, HypK has a strong stabilizing effect and inhibits the catalytic activity of Naa10 (refs. 19,20). Moreover, HypK binding to NatA induces structural rearrangements in Naa15, which allosterically inhibits Naa50 binding[25]. HypK has been reported to co-localize with polysomes[24] and contact nascent chains[26], while loss of HypK negatively impacts NatA function[24,27,28]. While the mechanisms of in vitro NatA inhibition by HypK are well characterized[19,20,25], the molecular details of NatA regulation in vivo remain vague. The C-terminus of HypK, including the UBA domain, is homologous to the C-terminus of NACα. However, the precise functional link between these proteins remains to be elucidated.

Aside from ribosomal proteins and rRNA, the largely unstructured heterodimer NAC also shapes and influences the biochemical properties of the PTE. Its large surface provided by the four charged and extended protein termini might have far-reaching implications on other RAFs that associate to the tunnel exit during translation. Work on NAC has already highlighted its importance in SRP regulation[29–32], protein folding and proteostasis[33], as well as MAP1 coordination[13,34]. Its omnipresence in this early stage of co-translational protein maturation strongly implies an additional role in the regulation of acetylation.

In this work, we report single-particle cryo-EM structures of multienzyme assemblies for coordinated NME-NTA. These include the canonical NatA heterodimer (Naa10, Naa15) on the human 80S ribosome. Unlike yeast NatE, human NatA binds at a non-intrusive distal position which allows concurrent binding of either NAC-MAP1 or MAP2 as presented here, and also most other RAFs as NatB, RAC, SRP, Ebp1, or SEC61 without steric clash. At this position, NatA forms a dynamic ring-like assembly with MAP1 and NAC, which encompasses the PTE or binds adjacent to MAP2. In both scenarios, successive NME and NTA is possible, although the PTE distance to the active centers is different. Overall, our structures provide evidence that RAFs can arrange in dynamic multi-protein assemblies at the PTE independent of nascent chains, to allow for rapid and highly specific co-translational enzymatic processing on the human ribosome.

## Results

### NatA binds to the ribosome at a distal position

One major function of Naa15 is to facilitate the positioning of Naa10 near the PTE during translation[23,35–37]. To examine how this is accomplished at the human ribosome, we reconstituted the NatA-80S binary complex in vitro and subjected the sample to single particle cryo-EM analysis. Data processing revealed an inherently dynamic NatA interaction with the ribosome, where the heterodimer adopts a multitude of conformations, which severely limits the local resolution (Supplementary Fig. 1). Nonetheless, this preliminary structure revealed that, compared to yeast NatE[23], human NatA accommodates a completely different non-intrusive off-center position near the PTE, in the following denoted as 'distal' site, independent of a nascent chain substrate or other RAFs. Recently, we reported several structures of MAP2 (ref. 13) and of Ebp1[38] that share the MAP2 fold on the ribosome. The central placement of MAP2 on top of the PTE would not result in a steric clash with NatA at its distal position and might, therefore, allow a coincident occupation of the ribosome by these two sequentially active enzymes. As an attempt to stabilize NatA at the distal site, we added MAP2 to the NatA-80S complex and recorded a cryo-EM dataset (Supplementary Fig. 2 and Supplementary Table 1). Indeed, concurrent binding of human MAP2 constrained the conformational freedom of NatA and improved the local resolution up to ~4 Å, without interfering with the NatA distal position (Fig. 1a and Supplementary Figs. 2, 3). In the ternary NatA-MAP2-80S complex, neither MAP2 nor NatA binding is altered in the presence of the other factor, and there is no direct interaction between their structured domains (Fig. 1a). Therefore, this dataset was used for a detailed analysis of the NatA-80S interaction.

Consistent with the reported functions of Naa15, the two major contacts formed between NatA and the ribosomal surface are mediated by this adapter protein (interface of 1100 Å² calculated using PISA[39]). The first contact is formed by a long α-helix detached from the Naa15 scaffold (helix α34, residues 591–628) (Supplementary Fig. 4), which wedges in the widened major groove of a protruding kink-loop of expansion segment ES7L(A) (proximal and extended: nucleotides 467-471 and 683-688) (Fig. 2a) and also bridges a large gap of over 55 Å to ES44L (Fig. 2b). The second contact to the PTE is formed by the N-terminal TPR motif (helices α1 and α2) positioned in between exposed ribosomal 28S rRNA helices (H19 and H24) and the peripheral lining of uL24 (Fig. 2c). While the local resolution is not sufficient to model specific side-chain interactions, it is clear that the positively charged surfaces of helix α34 and TPR1 contact the negatively charged rRNA backbone, which is also supported by an Alphafold 3 prediction (Supplementary Fig. 5)[40]. 3D classification revealed that NatA can still assume a range of conformations despite the presence of MAP2 (Supplementary Fig. 6). Long flexible linkers of helix α34 (16 N-terminal residues: 573–588, and 24 C-terminal residues: 629–652) that connect the helix to the Naa15 TPR-scaffold (Figs. 1a, 2) allow NatA to rappel further down towards the PTE while retaining all its ribosomal contacts.

While Naa15 contributes most prominently to the ribosome interactions, Naa10 also contacts uL24 in some of the NatA states that we captured (Fig. 2d) (interface less than 100 Å² calculated using PISA[39]). These interactions mainly involve uL24 helix α3 and an extended loop of Naa10. However, these contacts appear transient as NatA rotates in the distal site and are resolved with a local resolution of ~7 Å. Compared to the crystal structures of NatA, the Naa15 solenoid undergoes rearrangements in the first two TPRs, when adapting to the distal site. Similar rearrangements have also been observed through the stabilizing interaction with HypK[20,25]. Of note, the bending of the first two Naa15 TPRs occurs in opposite directions upon HypK or ribosome binding, indicating structural plasticity of the solenoid (Supplementary Fig. 7).

While the same structural features of Naa15 are also employed in ribosome binding of ScNatE[23], HsNatA uses the two contact sites to

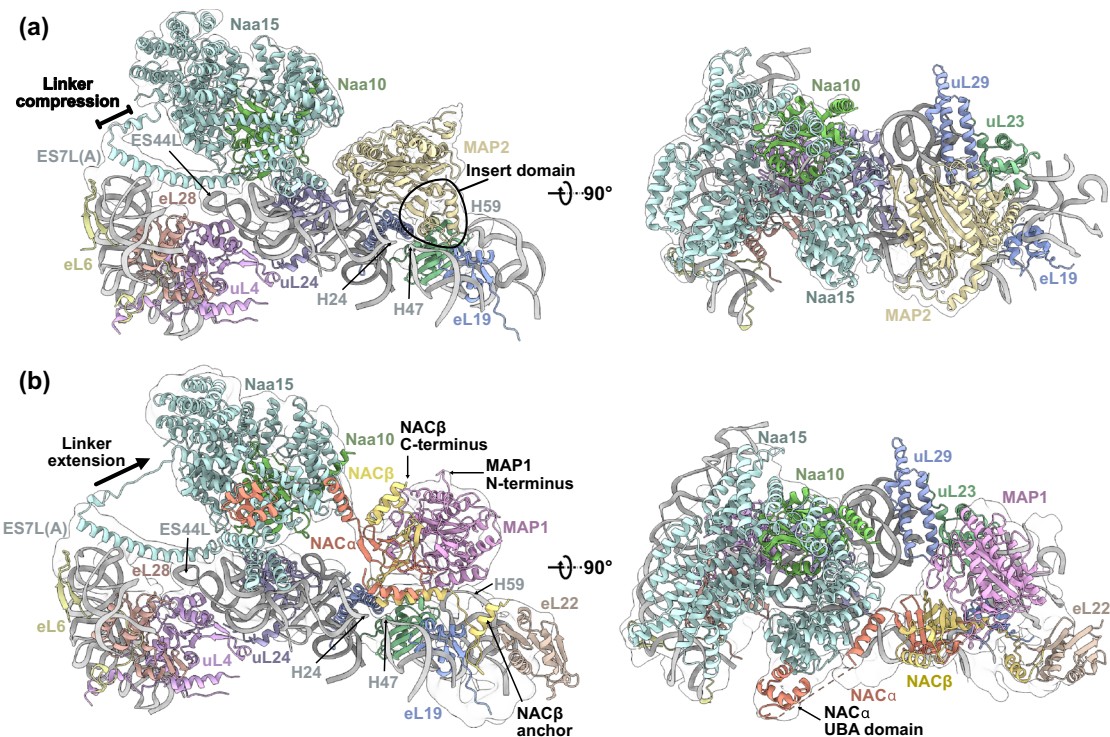

**Fig. 1 | Multi-NME-NTA enzyme assemblies on the human 80S ribosome. a** The ternary NatA-MAP2-80S complex around the PTE (interacting ribosomal components are shown and labeled). NatA is located at a distal site, while MAP2 binds independently and is positioned centrally on the PTE (approximately at the H24 tip). NatA binding to the distal site is maintained by the Naa15 TPR-solenoid exposing a long helix flexibly tethered to 28S rRNA expansions ES7L and ES44L. MAP2 interaction with the PTE is mainly mediated by the insert domain, which remodels 28S rRNA helix H59 as described before[12]. **b** The quaternary NatA-NAC-

MAP1-80S complex (same views as in Fig. 1a). The heterodimeric NACαβ chaperone is necessary to mediate the interaction between NatA and MAP1. Overall, the NatA complex is rotated further down towards the PTE and the flexible linker is more extended while the main ribosomal contacts are maintained, as shown in Fig. 1a. NAC uses its N-terminal NACβ anchor to interact with eL19 and eL22 as before[13] and its C-terminus points towards the N-terminus of MAP1. The UBA domain of NACα binds the Naa15 solenoid. The contours of the cryo-EM map are shown around the models (transparent).

bind in a completely different position off-center from the PTE, denoted here as the distal site (Fig. 3). Without the steric constraint formed by Naa50 in the *Sc*NatE structure, *Hs*NatA binding becomes even more dynamic. When superimposing the cryo-EM structure of *Hs*NatE[25] onto the distal position of *Hs*NatA on the ribosome, Naa50 is pointing far away from the tunnel exit. In this position it is completely solvent-exposed and would not contact the ribosomal surface (Supplementary Fig. 8), indicating that human NatE might have a different binding site than NatA.

As described in detail before[12,38], the strong binding of MAP2 on top of the PTE and the adaptable rRNA helix 59 is mainly due to the MAP2-specific insertion domain not present in the MAP1 family (Supplementary Fig. 4). In our previous cryo-EM structure of the human MAP2-80S complex[12], the B-arm of the long 28S rRNA expansion ES27L was recruited on top of the PTE by the N-terminus of MAP2 in a subset of particles (ES27L_out position, for definition, see ref. 41) (Supplementary Fig. 9). The structure of the ternary NatA-MAP2-80S complex now shows ES27L locked away in the ES27L_in position at the 60S/40S interface (as also observed in our binary NatA-80S complex). The absence of the MAP2-ES27L contact does not influence the MAP2 position on the PTE. From our data, it is not clear why NatA binding would influence the ES27L recruitment of MAP2. In the absence of MAP2 or in complex with NAC and MAP1, NatA decorated ribosomes also reveal ES27L at the 'in' position.

### NAC mediates the interaction between NatA and Map1

The ternary NatA-MAP2-80S assembly suggests a possible way of coordinating NME with NTA on the ribosomal PTE. However, in eukaryotes, a majority of NatA substrates are produced by MAP1.

from the lacking insert domain in MAP1, the two enzymes mainly differ in the architecture of their N-terminal extensions[11]. While the MAP2 N-terminus is unstructured and comprises segmented regions of alternatingly charged residues, the MAP1 extension harbors zinc-finger motifs and is implicated in NACβ binding[13]. However, the active center of the two enzymes is highly conserved, and their substrate specificity shows small differences[42]. To assess whether MAP1 and NatA can bind together on the ribosome to allow for successive NME and NTA, we mixed recombinant protein with purified human ribosomes for cryo-EM analysis. After extensive local classification of a preliminary dataset in cryoSPARC, we identified MAP1 and NatA on the ribosome in complex with endogenous NAC, which had co-purified with the 80S monosomes despite the high-salt condition (500 mM KOAc) during gradient centrifugation. To enrich NAC on the ribosome, we cloned, expressed, and purified NAC from insect cells and reconstituted the quaternary NatA-NAC-MAP1-80S assembly for cryo-EM data collection (Supplementary Fig. 10 and Supplementary Table 1). Several rounds of focused 3D classification were required to deal with the extensive heterogeneity that results from this mix of factors on the ribosome. NatA, NAC, and MAP1 all exhibit continuous flexibility and explore a range of different conformations. Together, these factors assemble into a dynamic ring-like arrangement that engulfs the tunnel exit, leaving a large lateral gap towards uL29 (Fig. 1b). Notably, Alphafold 3 (ref. 40) accurately predicts the differences between the HypK-NatA and NACα-NatA interaction, despite the similarity between the two NatA interacting regions (Supplementary Fig. 5). Sandwiched in between NatA and MAP1, NAC is positioned on top of 28S rRNA helices H24 and H47, in the same position where it has been found in the absence of other factors[29]. Like NatA, NAC does not require a nascent

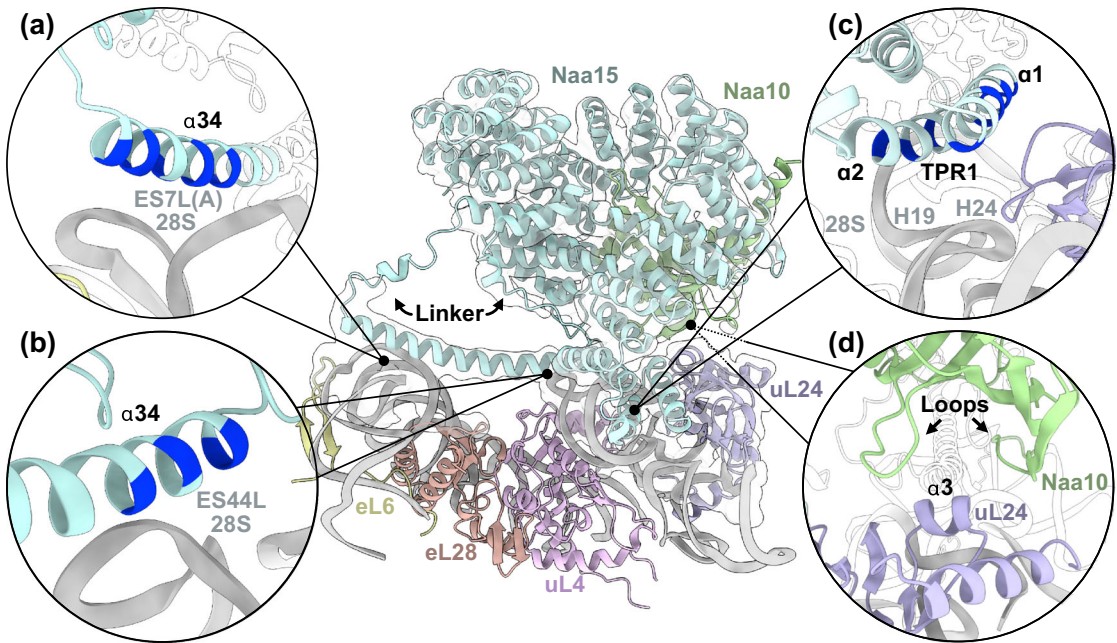

**Fig. 2 | Structure of NatA at the distal site.** NatA binding to the peptide tunnel exit (PTE) is shown for the ternary NatA-MAP2-80S assembly, with the cryo-EM map around NatA indicated in the central panel. The long and flexibly linked ribosome interacting Naa15 helix (α34) anchors the NatA complex via electrostatic interactions mainly with the phosphoribose backbone at kink-loops of (**a**) ES7L(A) and (**b**) ES44L. **c** The first Naa15 TPR1 (helices α1 and α2) touches down on the ribosomal surface in a hinging movement on top of 28S rRNA helices H19 and H24 in proximity to uL24 (violet). Again, contacts involve strong electrostatic interactions. The positions of positively charged residues are highlighted in a darker shade of blue. **d** In addition, Naa10 engages in transient interactions with helix α3 of uL24 (labeled), involving two extended loop regions.

chain to bind to the ribosome and to position itself at the PTE. NAC forms a peculiar heterodimer (α and β subunit of 215 and 206 residues, respectively) that assembles on a β-barrel core, flanked by two α-helical pairs and long flexible hydra-like termini. Situated at the PTE, the NAC dimerization domain packs tightly against the lateral side of MAP1 with the β-barrel-core placed almost perpendicular to the MAP1 active site, without blocking substrate entry.

The long terminal extensions of NAC can act over longer distances to facilitate various functions[13,29,43–45]. When NAC is positioned at the PTE, the N-termini of NACα and NACβ point towards the ribosome. The N-terminus of NACβ is resolved up to the short anchoring helix that facilitates NAC recruitment to the ribosome by binding next to eL19 and eL22 (refs. 29,44) (Fig. 1b). The NACα N-terminus is not visible in our reconstruction, but points towards the Naa15 interface. A potential role of this extension in the regulation of ribosome binding has previously been reported[34,43]. The unstructured C-terminal extension of NACβ has been shown to mediate MAP1 recruitment via a conserved hydrophobic patch that binds the N-terminal extension of the aminopeptidase[13]. While this interaction is not visible in our reconstruction, the trajectory of the NACβ C-terminus clearly points towards the N-terminus of MAP1. Taken together, our data show that the presence of NatA and the absence of nascent chains does not change MAP1-NAC binding at the PTE. Likewise, NatA binding is not per se affected by the presence of MAP1-NAC or MAP2.

**NACα and HypK employ conserved elements for specific interactions with Naa15**

In contrast to the unstructured NACβ extension, NACα includes an additional UBA domain at the very C-terminus (Supplementary Fig. 4). UBA domains are protein-protein interaction modules folded in a three-helix bundle and are found in e.g., the ubiquitin/proteasome pathway[46]. In the presence of NatA, NACα-UBA touches down on the Naa15 N-terminal helices (helices α4, α5, and α6) (Fig. 1b). The NACα-UBA/Naa15 interface involves a hydrophobic core surrounded by polar

and charged interactions (Supplementary Fig. 11). While the resolution is not high enough to build side-chain interactions (Supplementary Figs. 10, 12), the orientation of the UBA domain is clearly visible and further supported by the NatA-NAC Alphafold 3 prediction[40] (Supplementary Fig. 5).

In context of a nascent chain "handover" from NAC to the signal recognition particle (SRP), responsible for co-translational protein targeting to the ER membrane, NACα-UBA was found laterally attached to the SRP key-player SRP54 (ref. 29) (Supplementary Fig. 11). This shows that NACα-UBA can engage with SRP in protein targeting or NatA for NTA of a nascent chain by a conserved interface.

For NatA interaction, NACα employs contacts in addition to the UBA domain (Fig. 4a). The helix that connects the β-barrel-core to the UBA domain (denoted here as NACα contact helix (NACα−cH)) aligns outside Naa15 TPR4-TPR5 (helices α8, α9 and α10). This interaction is visible at ~7 Å (Supplementary Fig. 12) and forms around an aromatic cluster (Fig. 4b). Together, the NACα-cH and NACα-UBA share an interface of almost 1,000 Å² with Naa15 (PISA server[39]). Intriguingly, the NatA modulator HypK displays homology to NACα and also contains a UBA domain that is preceded by an α-helix which employs the same interface with Naa15 TPR4 to TPR6 (refs. 19,20,25) (Fig. 4c and Supplementary Fig. 4).

For NACα, however, the connecting cH is much shorter than the corresponding HypK-cH. This structural difference apparently contributes to a change in UBA position at the Naa15 solenoid. While NACα-UBA contacts the Naa15 N-terminal TPR2 and TPR3, HypK-UBA binds diametrically opposed at the Naa15 C-terminal TPRs. The function of the Naa15 solenoid as a binding hub for external helical elements is even further corroborated by our finding that the Naa10 C-terminal helix also crawls along Naa15 TPR6 and TPR7 in a highly similar fashion as NACα-cH (Supplementary Fig. 13).

Overall, the quaternary assembly of NatA, NAC, and MAP1 on the 80S ribosome is highly dynamic. Despite the numerous contacts that NatA forms to the ribosome and to NACα, it is still free to rotate in the

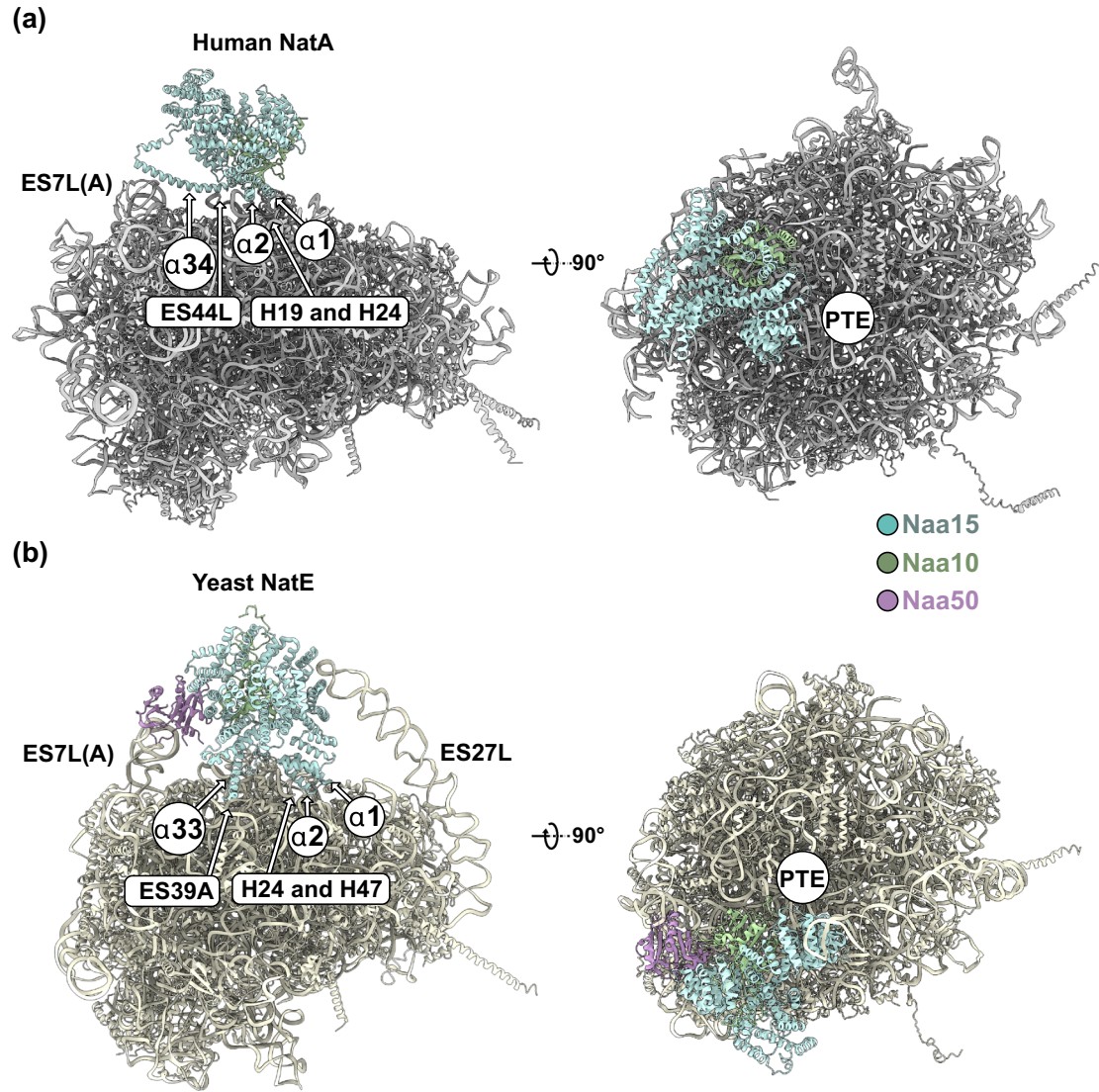

**Fig. 3 | Structural comparison of the human NatA and yeast NatE ribosomal binding sites. a** Binding of human NatA to the ribosome relies only on two contact points. The first contact is mediated by the first two N-terminal helices α1 and α2, which touch down on rRNA helices H19 and H24. The second contact is mediated by the long anchoring helix α34, which wedges in between ES7L(A) and spans across the ribosomal surface to also interact with ES44L. The human NatA complex is positioned further away from the peptide tunnel exit (PTE) (compare right panels). **b** Budding yeast NatE (PDB: 6HD7)[23] has four contact points with the 60S subunit of the ribosome. With contacts formed by helix α2, Naa15 is placed onto rRNA helices H24 and H47 directly adjacent to the PTE opening. The long anchoring helix α33 (corresponds to α34 in humans) makes a second contact with ES39A. Mediated by Naa50, the yeast NatE complex contacts ES7L(A). An additional contact is formed by the apex of ES27L and Naa15.

distal site while retaining all of its contacts (Supplementary Fig. 14). Compared to our NatA-MAP2-80S structure, the NatA complex is rotated 25° down towards the PTE and the first TPRs are shifted further away from uL24 (Supplementary Fig. 15), which is likely a consequence of the direct interaction with NACα. Throughout this motion, the N-terminal linker (residue 571–588) can extend and compress by one-third (between 38 and 50 Å) to allow the dynamic rotation of Naa15.

## Discussion

Nine different enzymes, namely two MAPs, two NMTs and five NATs need to be coordinated at the PTE in the context of a rapidly translating ribosome[1,2]. Once the nascent chain starts to emerge from the ribosome, the first two residues can become targets for modification by this elaborate pool of RAFs. While the nascent chain is still short and barely protruding from the PTE, these target residues are still in a predictable and accessible position. To exploit this narrow window of opportunity and to avoid unproductive overcrowding at the PTE, enzymatic processes necessary for the maturation of a growing nascent chain need to be well coordinated. NME by MAP1 or MAP2 and subsequent NTA by NatA are the most widely executed co-translational modifications in human cells[4]. Given the relative underrepresentation of these enzymes compared to the ribosome, their binding, activity, and eventual release must be tightly controlled.

In this study we show that NatA can bind the ribosome in a non-intrusive off-center distal site. While ribosome binding of many RAFs is mutually exclusive, the NatA distal site does not interfere with binding of most other known RAFs at the PTE, namely MAP1, MAP2, Ebp1, NatB, NAC, RAC, SRP, and SEC61 (Fig. 5a). A coincident binding of NatA with factors that function up- or downstream of acetylation, might allow for a more coordinated handover of the nascent chain substrate in between the subsequent maturation steps. To explore two different routes that might enable the coordinated handover between NME and NTA, we determined two cryo-EM structures of NatA assemblies,

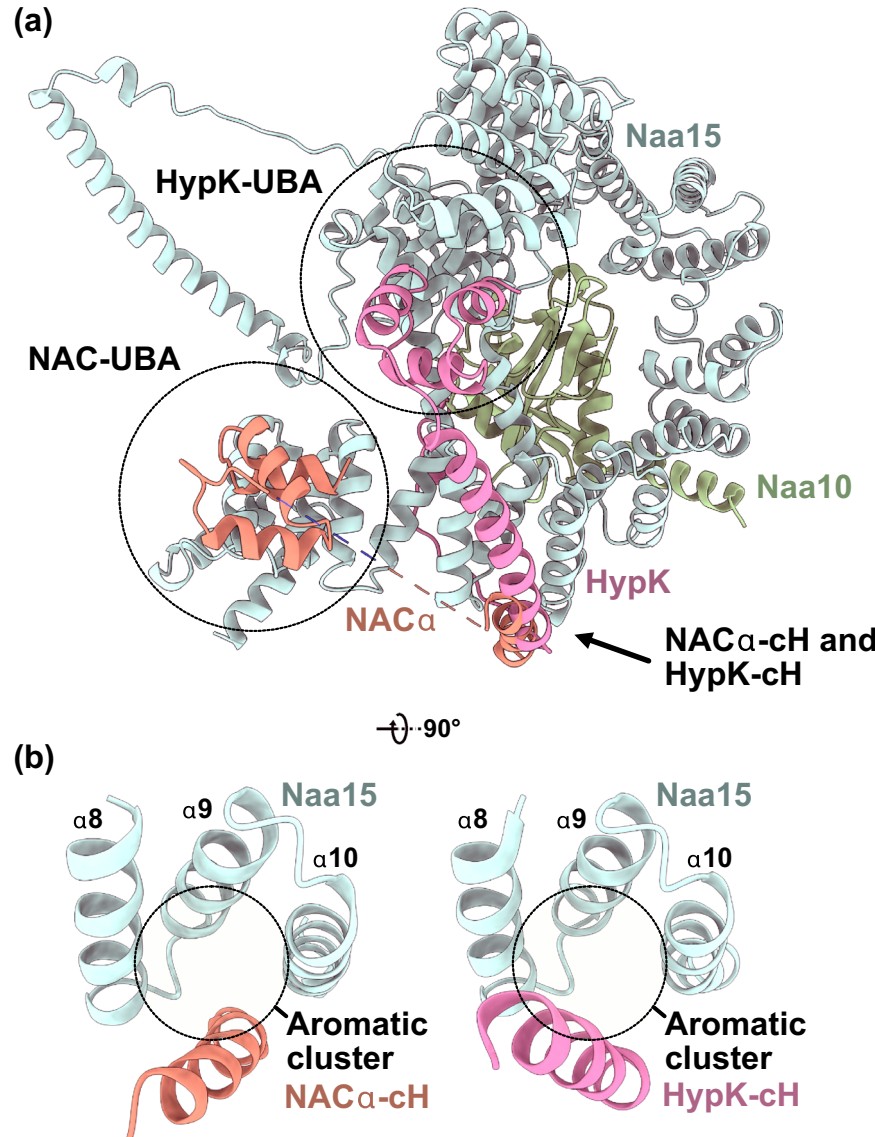

**Fig. 4 | HypK and NACα employ homologous elements for Naa15 interaction.**
**a** Superposition of NatA-NACα (C-terminus of NACα including NACα-contact helix (cH) and ubiquitin associated (UBA) domain, this study) in the quaternary complex with the NatA-HypK complex (pdb-ID 6C95 (ref. 19)). NACα-cH and HypK-cH both run along the surface of Naa15 helices α8 to α10. As the HypK-cH is longer than NACα-cH, the HypK-UBA domain is positioned near the C-terminal helices of Naa15 while the NACα-UBA domain is placed on top of helices α4 to α6 near the N-terminus of the Naa15 solenoid. **b** NACα-cH aligns with helices α8 to α10 of Naa15 around an aromatic cluster, in the same position as the HypK-cH.

indicating two possible routes of maturation with either MAP1 or MAP2.

In the ternary NatA-MAP2-80S structure, the individual binding sites of NatA and MAP2 were not affected by the presence of the other factor (Fig. 5b). However, the rotational freedom of NatA was limited, due to the PTE occupation by MAP2, which allowed for a higher-resolution reconstruction of NatA in the distal site. In our previous cryo-EM structures of the binary MAP2-80S complex, we reported that the enzyme can undergo a dynamic rotation around the insert domain in the presence of nascent chains[12].

Such a rotation of MAP2 might allow NatA to rappel further down toward the PTE and position the Naa10 catalytic subunit closer to the emerging nascent chain. The motion of NatA towards MAP2 might also cause displacement of the latter, to make room for binding of the subsequent processing factor. Within the ternary NatA-MAP2-80S assembly, an emerging nascent chain would need to bridge a gap of ~40 Å and ~75 Å to reach the active sites of MAP2 and Naa10,

respectively (measured from 28S rRNA G2416 (ref. 3)) (Fig. 5b). For the structure determination of NatA decorated ribosomes we used translationally silent monosomes to elucidate how these enzyme complexes compile before a nascent chain emerges. Additional data on programmed ribosomes might help to understand how these multi-factor complexes react to an emerging substrate.

While the interaction between the NACβ C-terminus and MAP1 is not visible in our reconstruction, the interaction between NACα and NatA is well defined (Fig. 5c). A similar interaction has also been described for the NatA-HypK complex, where NACα homologue HypK also contacts Naa15 in parallel to helices α8 to α10[19,20,25]. To date, the precise function of HypK is not understood. In our structures, we observe that structural adaptations in the first two Naa15 TPRs (helices α1 to α4) occur when NatA adopts the distal position on the ribosome (Supplementary Fig. 7). Interestingly, HypK binding also induces a similar bending of these helices, however, in opposing direction[19,20]. The stabilizing effect of HypK binding on NatA has been well

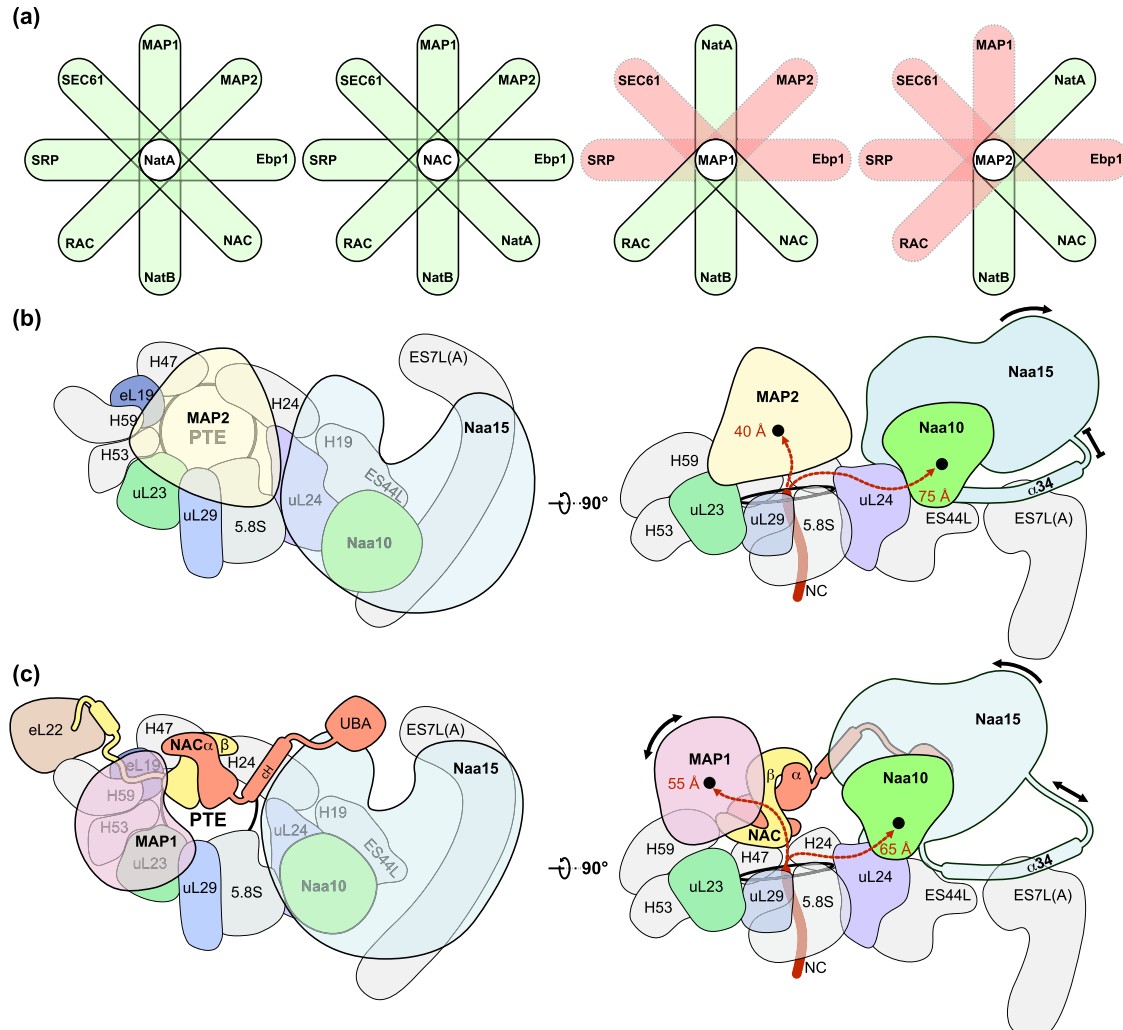

**Fig. 5 | Two coordinated multi-factor assemblies compile on the PTE. a** RAF combinations centered on the main players of this study that could form at the PTE without steric interference are shown in Venn diagram-like representations. RAFs include MAP1 (this study), MAP2 (ref. 12), NatA (this study), Ebp1 (ref. 38), NAC (this study), NatB[48], RAC[57], SRP[29], and SEC61 (ref. 58). The NatA off-center distal position and the non-intrusive NACβ anchoring position does not interfere with binding of any of these factors (highlighted in green). In contrast, MAP1 would clash with MAP2, Ebp1, SRP, and SEC61, while MAP2 would clash with MAP1, Ebp1, RAC, SRP, and SEC61. Factor combinations that are not possible due to steric clash are highlighted in red. **b** Schematic top and side view of the ternary NatA-MAP2–80S assembly. MAP2 is positioned centrally on the peptide tunnel exit (PTE) next to NatA. NatA is anchored to ES7L(A) and ES44L with its α34 helix, and the connecting

linker is compressed. In this assembly, the distance of an emerging nascent chain (NC) to the active sites of MAP2 and Naa10 are 40 Å and 75 Å, respectively (measured from the last 28S rRNA PTE constriction at G2416 (ref. 3). **c** Schematic top and side view of the quaternary NatA-NAC-MAP1-80S assembly (views as in Fig. 6b). NAC is anchored with its NACβ helix adjacent to eL22 while the dimerization domain is placed on top of rRNA helices H24 and H47. The NACα-contact helix (cH) and ubiquitin-associated (UBA) domain contact Naa15, and the dimerization domain is in close contact with MAP1. The NatA complex is still anchored on ES7L(A) and ES44L via helix α34, but the connecting linkers are almost completely extended. The distance to the active site of MAP1 and Naa10 would be 55 and 65 Å for the two enzymes, respectively. Both NatA and MAP1 remain dynamic within the assembly and can rotate towards and away from the PTE as indicated.

described[19,20]. Rigidification of the Naa15 scaffold by HypK binding might prevent the distal site interaction and restrict its structural plasticity, unless HypK rearranges in some way. These observations would corroborate the proposed function of HypK as a NatA inhibitor when not bound to the ribosome[19,20,25]. Indeed, reports that HypK contacts nascent chains and co-localizes with polysomes contradict the inhibitory function on ribosome binding[24,26]. NatA has the broadest substrate specificity of any N-acetyltransferase[4] and could potentially co-occupy the PTE with most other RAFs. Furthermore, the Naa15 scaffold constitutes a central binding hub for a multitude of factors (e.g., Naa10, Naa50, HypK, and NACα). This plethora of possible Naa15 interactions needs to be regulated in some way. Perhaps HypK acts as a 'specificity filter' that ensures that NatA is only employed on the ribosome in the right condition, possibly in the presence of other

factors that can induce rearrangements in HypK. Herein, NAC would be a prime candidate, as its NACα−cH utilizes the same interaction surface on Naa15 as the HypK-cH. Given the low nanomolar affinity of HypK to the NatA complex[19,20] it has been suggested that in human cells, these proteins form an obligate ternary complex in vivo[19]. In our NatA-MAP2-80S structure, we observed no direct interaction between NatA and MAP2. If and how HypK might be displaced in the presence of MAP2 remains to be explored.

The diverse functions inherent to the terminal extensions of NAC are only partially understood. The precise function of the ~70 residue long NACα N-terminal extension remains to be described. Interestingly, the 'QQAQLAAAE' motif in the NACα-cH, which mediates the Naa15 interaction described here, is duplicated in the N-terminal extension of NACα ("QQAQLAAAAE"). In the context of NatA,

Alphafold 3 (ref. 40) predicts this region to form a helix (Supplementary Fig. 5). Whether the N-terminus of NACα can also contact Naa15 remains to be explored. Connected to the NACα-cH via a short unstructured linker, NAC possesses a UBA domain that touches down on the N-terminal helices of Naa15. In complex with SRP, NACα-UBA was found to also contact SRP54 (ref. 29). This interaction is similar to the NACα-Naa15 interaction, as the UBA domain with its exposed hydrophobic patch is positioned on two parallel helices (albeit at different angles) (Supplementary Fig. 11). Most SRP targets are not subject to NME and NTA by MAP1 and NatA while aberrant acetylation of such nascent chains has been shown to inhibit translocation[47]. The coordination of NME and NTA by NAC or the coordination of SRP recruitment by NAC are, therefore, likely two mutually exclusive processes.

Most RAF interactions with the ribosome can occur independent of a nascent chain substrate (e.g., MAP2 (ref. 12), Ebp1 (ref. 38), NAC, NatA, NatB[48]). Furthermore, all RAF enzymes only target the first two residues of the nascent chain. These residues will be most accessible while nascent chains are still short (e.g., <50–60 residues). Given the high speed of translation, in addition to the flexibility and variability of nascent chains, there would likely not be enough time to wait for the N-terminal residues to emerge from the PTE before the enzymes associate to the ribosome. Instead, we postulate that specific multiprotein 'starter-kits' form on the PTE to reside in a reactive state and await emerging substrates. Once the nascent chain starts to grow beyond the PTE, it might induce conformational changes in these RAFs (e.g. a rotation of MAP2 (ref. 12)) that allow the recruitment of downstream factors. A 'starter-kit' composed of MAP1, NatA, and NAC would combine the functionalities of NME, NTA, and molecular chaperoning. Perhaps, one alternative 'starter-kit' might compose of RAC, NatA, and MAP1. A major question that remains is how the composition of these start assemblies could be controlled. Localized translation of a specific subset of mRNAs in cytoplasmic condensates is an attractive idea[49] and might result in an enrichment of certain RAFs with ribosomes. This could be an elegant way to regulate local translation efficiently despite the low abundance of many RAFs. Taken together, our study provides insights into how multi-protein assemblies orchestrate the first two steps of co-translational enzymatic processing on the human ribosome. Understanding the principles of the fundamental biological processes of NME and NTA and their integration in the regulation of protein biogenesis in human cells and in all domains of life is still a major task.

## Methods

### Sample preparation

Genes encoding *Hs*NACα, *Hs*NACβ, and *Hs*Naa10 were purchased from IDT peptide. Genes encoding *Hs*Naa15 and *Hs*MAP1 were subcloned from vectors *pFastBac-HIS-Hsnaa15* and *pET24d-HIS-TEV-Hsmap1*, respectively. *HsnatA* was cloned into *pFastBacDuet* plasmid with *naa10* expressed under the polyhedrin promotor with a 3C-protease cleavable C-terminal HIS-tag (*naa10-GSGS-3C-GSGS-10HIS*). Untagged *naa15* was co-expressed under the p10 promotor. N-terminally HIS-tagged *map1* was expressed under the polyhedrin promotor from *pFastBacDuet* plasmid with a 3C-protease recognition site between the HIS-tag and *map1* (*10HIS-GSGS-3C-GSGS-map1*). *Nac* was also cloned into a *pFastBacDuet* plasmid with *nacα* expressed under the polyhedrin promotor with an N-terminal 3C-cleavable Strep-Tag II (*Strep-II-GSGS-3C-GSGS-nacα*). *Nacβ* was co-expressed from the p10 promotor without an affinity tag. The *Hsmap2* expression vector was previously generated[12] by subcloning *Hsmap2* from a *pET24d-6HIS-GSGS-Hsmap2* plasmid into expression vector *pFastBacDuet-10HIS-GSGS-3C-GSGS-Hsmap2*. Oligonucleotides used for cloning are listed in Supplementary Table 2.

All proteins were expressed in Sf9 cells (Cat. No. 12659017). Briefly, insect cells were grown at 27 °C and 80 rpm to be infected at a density of $2 \times 10^6$ cells/ml. Expression was continued for 72 h, and cells were harvested by centrifugation at $1500 \times g$. The pellet was washed once with PBS, flash-frozen in liquid nitrogen, and stored at −80 °C until further use. For purification, pellets were lysed in a microfluidizer in the presence of $1 \times$ protease inhibitor (Roche) and cleared by ultracentrifugation at $50,000 \times g$ for 20 min. Cleared lysate was passed through a 0.2 μm syringe filter, and proteins were purified at 4 °C. The purification of MAP1, MAP2, NAC, and NatA were done by a two-step purification scheme. To produce pure NAC, *Strep-II-GSGS-3C-GSGS-nacα*, and *nacβ* were co-expressed in Sf9 cells. The recombinant protein was captured on StrepTactin™ Agarose (Purecube). To obtain pure NatA, *naa10-GSGS-3C-GSGS-10HIS*, and *naa15* were co-expressed in insect cells. Naa10 was captured on $Ni^{2+}$- Agarose beads (Qiagen). 10HIS-GSGS-3C-GSGS-MAP1 and 10HIS-GSGS-3C-GSGS-MAP2 were also purified on $Ni^{2+}$- Agarose beads. After the capture step, NAC, NatA, MAP1 and MAP2 purification were done by the same protocol. Affinity tags were cleaved off with 3C-protease, and eluted protein was concentrated to 10 mg/ml. To obtain stoichiometric complexes and pure MAP1 and MAP2, samples were subjected to SEC on a S200 16/600 (Cytiva) column in SEC buffer (20 mM HEPES KOH, 150 mM KOAc, 5 mM $MgOAc_2$, 1 mM TCEP, pH 7.4). The purity of proteins was evaluated by SDS-PAGE and analytical SEC on S200 increase 3.2/300 GL (Cytiva) (Supplementary Fig. 16).

Human 80S ribosomes were isolated from HeLa S3 cells (Cat. No. 87110901) as described[38], with the sucrose cushion adjusted to a KOAc concentration of 500 mM. For cryo-EM analysis of the NatA-80S complex, ribosomes were subjected to SEC (20 mM HEPES KOH, 600 mM KOAc, 5 mM $MgOAc_2$, 1 mM TCEP, pH 7.4) on a Superose 6 10/300 column (Cytiva) to remove any co-purified RAFs from the PTE. Afterward, the buffer was adjusted to 20 mM HEPES KOH, 150 mM KOAc, 5 mM $MgOAc_2$, 1 mM TCEP, and pH 7.4. For cryo-EM datasets of NatA-NAC-MAP1-80S, ribosomes were not subjected to high-salt SEC to additionally retain endogenous co-purified NAC. After purification, ribosomes were flash-frozen in liquid nitrogen and stored at −80 °C until further use.

### Cryo-EM grid preparation

Before sample vitrification, copper grids were glow-discharged in the Solarus plasma cleaner (Gatan, Inc.) for 60 s in an oxygen atmosphere. For the data acquisition of NatA, 500 nM of high-salt washed 80S ribosomes were mixed with 12.67 μM of purified NatA. For the complex of MAP2 and NatA, 500 nM of high-salt washed 80S ribosomes were incubated with 30 μM of MAP2 and 30 μM of NatA for 30 min at room temperature. The sample preparation for the NatA-NAC-MAP1-80S complex was performed at 37 °C. Here, 500 nM of 80S ribosomes (not high-salt cleaned) were incubated with 5 μM of NAC for 10 min at 37 °C. NatA and MAP1 were added one after the other, with 10 min incubations in between, each to a final concentration of 15 μM. From each sample, 3 μl were frozen in liquid ethane with the Vitrobot Mark IV (FEI company) on 2/1 Quantifoil Multi A holey carbon supported grids (Quantifoil, Multi A, 400 mesh). Freezing was carried out with Whatman #1 filter papers at 4 °C with a blot force of 0, 10 s wait time, and 100% humidity. Grids were stored in liquid nitrogen until data acquisition.

### Data collection

Two datasets of the NatA-80S sample were collected on a Glacios transmission electron microscope (Thermo Fisher Scientific) operated at 200 keV with the Falcon 3 direct electron detector (Thermo Fisher Scientific) that collected at a pixel size of 1.223 Å/pixel and a magnification of 120,000. The total dose per micrograph was 51.65 and 55.07 e⁻/Å² for the two datasets, respectively. Three datasets of the NatA-NAC-MAP1-80S assembly were also collected on the Glacios transmission electron microscope at a magnification of 120,000 and a pixel size of 1.223 Å/pixel. The total dose for the three datasets was

53.50, 53.97, and 53.96 e⁻/Å², respectively. The dataset of NatA-MAP2-80S was collected at ESRF on a Titan Krios electron microscope (Thermo Fisher Scientific, FEI Company) on a K3 Summit direct electron detector (Gatan, Inc.) that collected movies at a pixel size of 0.84 Å/pixel at a magnification of 105,000. The total dose of the NatA-MAP2-80S dataset was 41.28 e⁻/Å². The data acquisitions were set up and monitored with EPU (Thermo Fisher Scientific).

## Data processing
Detailed descriptions of the processing schemes for all datasets can be found in the Supplementary information. Briefly, movies were imported and pre-processed in CryoSPARC[50], including Patch Motion Correction and Patch CTF Estimation. Ribosomes were picked with the Blob Picker and extracted for 2–3 subsequent rounds of 2D classification. Selected particles were subjected to ab-initio reconstruction into three classes, followed by Heterogeneous refinement. Classes containing 80S ribosomes were pooled and refined using the Homogeneous Refinement job in CryoSPARC. The remaining particles were subjected to several rounds of focused 3D Classification with different masks to subsequently deal with the compositional and continuous heterogeneity of the factors at the PTE. After classification, masks were generated to subtract the 80S signal, with the Particle Subtraction job and local masks around the tunnel exit were used for final Local Refinements. Local Resolution Estimation jobs were run for global and local refinements.

## Cryo-EM model building, refinement, and analysis
The high-resolution cryo-EM structure of the human ribosome (PDB ID: 6ek0)[51] was used as a starting point for model building. For the building of NAC and NatA, Alphafold models were generated to also predict the flexible insertions and extensions[52]. For building the ternary NatA-MAP2-80S complex, our MAP2-80S structure was used as a starting model (PDB ID: 8ONY)[12]. For building the NatA-NAC-MAP1 interaction, the cryo-EM structure of NAC-MAP1-80S was used (PDB ID: 8P2K)[13]. For model building, the component proteins were first rigid body fitted into the cryo-EM map using UCSF Chimera-X[53]. Atomic model building was performed in Coot[54], and the preliminary model was refined and validated in PHENIX suite[55,56]. The refinement statistics for all cryo-EM datasets are shown in Supplementary Table 1.

## Alphafold 3 prediction
Alphafold 3 was used to predict the interaction between NAC and NatA on July 19 2024 (ref. 40). Unstructured regions of Naa15 (UniProt: Q9BXJ9), Naa10 (UniProt: P41227), NACα (UniProt: Q13765), and NACβ (UniProt: Q02642) were removed for the prediction to improve the readability of the confidence plots. Sequences used for structure prediction are available in the source data file.

## Figure preparation
Figures were prepared in UCSF Chimera-X[53].

## Reporting summary
Further information on research design is available in the Nature Portfolio Reporting Summary linked to this article.

## Data availability
All structural data that support the findings of this study have been deposited in the Protein Data Bank (PDB) (coordinates) and EMDB (maps) with the accession codes 9FPZ and EMD-50641 for the ternary NatA-MAP2-80S assembly and 9FQ0 and EMD-50642 for the quaternary NatA-NAC-MAP1-80S assembly. Of note: only the PTE is modeled for the human 80S ribosome. The entire human 80S ribosome can be superposed on PTE components (except 28S rRNA helix H59, which is flexible). For the analysis of results and making of figures, the following published structures were used: MAP2-80S (8ONY), SRP-NAC-80S (7QWQ), RAC-80S (7Z3N), NatE-80S (6HD7), NAC-MAP1-80S (8P2K), Ebp1-80S (6SXO), NatA-HypK (6C95), NatA (6C9M), NatE (6PPL), NatB-80S (8BIP), SEC61-80S (3J7Q), and Hs80S (6EK0). Source data are provided with this paper.

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

## Acknowledgements

We acknowledge the excellent technical support by Marina Pelzl, Britta Klem, and Astrid Hendricks. We acknowledge access to the infrastructure of the cryo-EM Network (HDCryoNET) at Heidelberg University, and support by Dirk Flemming (BZH), Jan Rheinberger (BZH), Lutz Nücker (BZH), and Götz Hofhaus (Bioquant). All cryo-EM grids and preliminary datasets, leading up to the final high-resolution datasets, were screened and acquired in our in-house facilities. We acknowledge the European Synchrotron Radiation Facility for the provision of beam time on CM01, and we would like to thank Michael Hons for assistance. Further, we acknowledge the services SDS@hd and bwHPC supported by

the Ministry of Science, Research and Arts Baden-Württemberg and the German Research Foundation through grants INST 35/1314-1 FUGG and INST 35/1134-1 FUGG. This work was supported by the Leibniz Program of the Deutsche Forschungsgemeinschaft to I.S. (SI 586-6).

## Author contributions

M.K., K.W., and I.S. designed the study, analyzed the data, and wrote the paper. M.K. generated all DNA constructs, purified proteins, and ribosomes. Cryo-EM grids were prepared by M.K., and data were acquired and processed by M.K. K.W. and M.K. build the atomic models.

## Funding

## Competing interests

The authors declare no competing interests.
