## [Peer Review File · Nature Communications]

Multi-protein assemblies orchestrate co-translational enzymatic processing on the human ribosomeREVIEWER COMMENTS

Reviewer #1 (Remarks to the Author):

Nascent chains are co-translationally modified during early stages of protein synthesis. Several enzymes that catalyze these modifications are tethered to the polypeptide exit tunnel on the large subunit, either by direct binding or by association with adaptors. While the different factors and their ability to bind to ribosomes are well-established, many details remain to be identified, including the molecular details of the ribosome binding, the interactions among these enzymes, mutual exclusive binding patterns and modes of cooperations of these factors. In the current study, Sinning and coworkers presented exciting structures of several complexes containing ribosomes and different sets of nascent chain-modifying enzymes. The study is very clear and the figures are beautiful. They present fascinating snapshots of different structures, which allow speculations about the dynamic recruitment of such enzymes during protein synthesis. The study also shows surprising differences between the human system (studied here) and baker's yeast (were studied before). The study clearly advances our understanding of the molecular interactions of factors binding to the ribosomal tunnel exit. Some minor points might be considered.

Minor points

1. Figure 1 compares structures containing MAP1 and MAP2. The authors propose from their model that the length of a helix in Naa15 determines which aminopeptidase is bound. In the extended state, MAP1 is bound and in the condensed state, Naa15 binds MAP2. However, from these two 'snapshots' it is not possible to deduce that the length of this linker really determines which co-factor is bound. The extended or non-extended linkers might rather be a structural consequence of the bound aminopeptidase without further physiological relevance. Mutants with stabilized shorter or longer linker domains could however clarify whether the linker lengths is the critical determinant in MAP recruitment.
2. Fig. 6 is important as it explains the conclusions of the study in a general way, highlighting which combinations of ribosome-bound factors can be present on the ribosome simultaneously. However, I found the presentation of the Venn-like diagrams in (a) confusing and the color code not intuitive. The authors should: first, show which factors can be present simultaneously on the ribosome, and second, come up with a speculative model on the dynamic recruitment of the different factors during translation. This will be of more relevance to most of the readers.
3. Figure 4 carries rather limited information. The authors should consider exchanging this figure with Extended Data Figure 6 which compares the human and yeast complexes. Of course, I also understand if the authors prefer to show their primary data in the main figures.

Reviewer #2 (Remarks to the Author):

The manuscript by Sinning and coworkers report on the cryo-EM structures of human NatA bound alone to the 80S ribosome, and together with Methionine Aminopeptidase 2 (MAP2) and MAP1 plus the Nascent Polypeptide Associated Complex (NAC), all in the absence of nascent chain assembly. A comparison of the structures reveals that (1) NatA binds dynamically to a 'distal' binding site on the ribosome that does not interfere with MAP1 or MAP2 binding nor with most other ribosome-associated factors (RAFs), (2) MAP2 binding appears to constrain the conformational freedom of NatA but binds in a position that appears to be incompatible with N-terminal acetylation as MAP2 binds on top of the Protein Tunnel Exit (PTE), (3) NatA binding appears to effect the positioning of the B-arm of the long 28S rRNA expansion ES27L (from the OUT to the IN position), (4) NAC binds at a position between NatA and MAP1, and although not visible in the current map, the NAC N-terminus points to the Naa15 subunit of NatA and the NAC C-terminus points towards MAP1 and is thus inferred to bridge communication between NatA and MAP1, (5) NAC and HYPK employ conserved UBA-like domain to contact distinct sites on the Naa15 subunit of NatA.

Although the current studies appear to map, and confirm previous studies, on the binding positions of NatA, NAC and MAP1 and 2 to the ribosome how these proteins coordinate NatA activity on the ribosome is not elucidated by these studies. Thus, the conceptual advance of the current study is limited.

Other major points are:

1. In addition, given that NatA appears to be dynamically bound to the ribosome with limited resolution and a clear preferred orientation issue with all the structures and particularly with the NatA/NAC/MAP1 structure (panel e of Extended Figure 8), it is unclear to me how the authors derive the detailed protein-protein interaction cartoon and stick models that are shown throughout the manuscript. Such protein-protein interactions needs to be supported by showing the corresponding models placed within the cryo-EM densities. Biochemical studies could also support the proposed model for NAC by NatA.

2. How was HYPK incorporated into the complex? This is not discussed. Was HYPK included in all to the NatA reconstructions?

3. While the study clearly reveals that NAC and HYPK bind to different surfaces of Naa15, how NAC relieves HYPK-mediated inhibition of NatA is not clear from this study.

4. The distinction that the authors make about a 'distal' NatA binding site is not clear. Is there an alternative 'proximal' site? This needs to be elaborated.

5. Pg 5, Can the authors rationalize why NatA is better ordered in the MAP2 containing complex than NatA alone if there are no visualized or proposed contacts between the two proteins?

6. It's not clear from the discussion or figures why the ES27Lout position is incompatible with NatA binding. Can this be better explained? The significance of the ES27L configuration on function is also unclear.

7. If endogenous NAC was already bound in the complex, why did the authors need to reconstitute with recombinant NAC for SPA cryo-EM?

Minor points

Pg 2, change "The zinc-finger motive containing" to "The zinc-finger motif containing"

Extended Data Fig. 3 should be rotated 90 degrees clockwise.

Reviewer #3 (Remarks to the Author):

As the nascent polypeptide chain emerges from the ribosomal tunnel, it is subject to a number of co-translational modifications, including N-terminal excision and acetylation, which are essential in higher eukaryotes. Despite decades of research, the interplay between the different enzymes that mediate these processes at the tunnel exit is not well-understood. The study of Klein et al reports structural insight into the interplay of MAP1, MAP2, NatA and NAC on the human 80S ribosome. The authors show that unlike yeast NatE, human NatA binds in a more distally located position at the tunnel exit that does not interfere with binding of other RAFs, which is illustrated by addition cryo-EM structures of NatA with NAC and MAP1, or MAP2. The manuscript is generally well-written and presented and the results will be of general interest to researchers in the fields of translation and co-translational protein folding/processing. I have two comments that should be addressed and a few minor points.

Comments.

1. The first comment is that it is a little disappointing that authors use vacant 80S ribosomes for their studies rather than making the effort to form more physiological functional complexes i.e. translating ribosomes with nascent polypeptide chains. After all, the factors under study are all involved in processing/targeting of the nascent polypeptide chain. It seems somewhat contradictory to highlight in the abstract that "...how these enzymatic processes are coordinated in the context of a rapidly translating ribosomes has remained elusive"...and then to talk about multi-enzyme assemblies on 80S ribosomes...as though these are in anyway related to translating ribosomes...At a minimum the authors should add "vacant human 80S ribosomes" and also mention this limitation clearly in their manuscript discussion.

2. The second comment is about the absence of non-segmented cryo-EM density images for the complexes in Figure 1. These should be added to this figure with the models underneath. The other point is that the average and local resolutions of the different factors in the complexes should be clearly states in the main text, rather than hidden away in the ED Figure file. There are local resolution images in the the ED however the authors need to add transparent density images with molecular models next to them so the reader can understand which regions of the maps have which resolution. The blobby looking density in Figure 3 does not instill confidence, but presumably these are filtered maps – but nothing is stated in the legend. Perhaps the authors should provide some density images to convince the reader that the sidechain interactions they describe, for example, in Figure 2, 4 and 5, are supported by their density and local resolution.

Some minor points:

1. Line 43 and 324, why is the initiator methionine in a highly predictable position. I could imagine it is highly flexible and the location is not able to be predicted at all.

2. Line 77, "as a the"

3. The colors for Naa15 and rRNA seem very similar and hard to distinguish. The authors should change this...perhaps grey as they used in their model in figure 6?

4. Line 241, "Alike"

REVIEWER COMMENTS

Reviewer #1 (Remarks to the Author):

Nascent chains are co-translationally modified during early stages of protein synthesis. Several enzymes that catalyze these modifications are tethered to the polypeptide exit tunnel on the large subunit, either by direct binding or by association with adaptors. While the different factors and their ability to bind to ribosomes are well-established, many details remain to be identified, including the molecular details of the ribosome binding, the interactions among these enzymes, mutual exclusive binding patterns and modes of cooperations of these factors. In the current study, Sinning and coworkers presented exciting structures of several complexes containing ribosomes and different sets of nascent chain-modifying enzymes. The study is very clear and the figures are beautiful. They present fascinating snapshots of different structures, which allow speculations about the dynamic recruitment of such enzymes during protein synthesis. The study also shows surprising differences between the human system (studied here) and baker's yeast (were studied before). The study clearly advances our understanding of the molecular interactions of factors binding to the ribosomal tunnel exit. Some minor points might be considered.

We thank reviewer #1 for the kind remarks and positive reception of our data and manuscript.

Minor points

1. Figure 1 compares structures containing MAP1 and MAP2. The authors propose from their model that the length of a helix in Naa15 determines which aminopeptidase is bound. In the extended state, MAP1 is bound and in the condensed state, Naa15 binds MAP2. However, from these two 'snapshots' it is not possible to deduce that the length of this linker really determines which co-factor is bound. The extended or non-extended linkers might rather be a structural consequence of the bound aminopeptidase without further physiological relevance. Mutants with stabilized shorter or longer linker domains could however clarify whether the linker lengths is the critical determinant in MAP recruitment.

We apologize for not being clear. We do not believe that the degree of linker extension determines whether NAC-MAP1 or MAP2 interacts with NatA, and we agree that the rotation of NatA is likely a consequence of the interaction with NAC. We do not want to cause confusion and have added a sentence to the results to clarify (last paragraph before the discussion).

2. Fig. 6 is important as it explains the conclusions of the study in a general way, highlighting which combinations of ribosome-bound factors can be present on the ribosome simultaneously. However, I found the presentation of the Venn-like diagrams in (a) confusing and the color code not intuitive. The authors should: first, show which factors can be present simultaneously on the ribosome, and second, come up with a speculative model on the dynamic recruitment of the different factors during translation. This will be of more relevance to most of the readers.

We chose this kind of Venn-like diagram for Fig. 6a because it only shows the possible 2-factor combinations that can assemble on the ribosome. The protein of interest is positioned centrally and all other individual proteins that could bind together with this central factor are shown in color. We also considered drawing up a more conventional Venn diagram that shows all possible factor combinations. However, upon experimenting with this idea, we found it too confusing because there are too many 2-factor, 3-factor and even 4-factor combinations that would be possible without steric clash.

Just a few examples of possible 3- and 4-factor combinations include:

- NatA-NAC-MAP1
- NatA-NAC-MAP2
- NatA-RAC-MAP1
- NatA-NAC-SRP
- NatA-NatB-NAC-MAP2
- NatA-NatB-NAC-MAP1

With the simpler Venn-like diagram we hope to show that NAC and NatA both have unique 'off-center' positions that allow simultaneous binding of most other ribosome associated factors.

We agree however, that Fig. 6a could be better refined to avoid confusion, especially regarding the color code. We have addressed this by a simpler 2-color code, with green indicating that simultaneous interaction is possible at the ribosome, and red indicating that this is not possible. We have changed the figure legend accordingly.

Regarding a speculative model for the dynamic recruitment of different factors during translation:

Co-translational enzymatic processing happens on a relatively short nascent chain quickly after it emerges from the ribosome. Given the high speed of translation, we are convinced that the factors that coordinate enzymatic processing assemble on the ribosome before a nascent substrate emerges. From the data presented here we do not want to speculate on the order of binding events that eventually form these multi-protein assemblies at the ribosome. Work on NAC and MAP1 (Gamerding 2023) has shown that MAP1 is recruited by NAC, indicating that the latter needs to be present at the ribosome first. We also believe that NAC will be present before NatA but do not show data here to provide evidence for this. However, we show that MAP2 binding can happen independent of NatA, and NatA can bind independent of MAP2. Here it is not clear which factor will be present first.

3. Figure 4 carries rather limited information. The authors should consider exchanging this figure with Extended Data Figure 6 which compares the human and yeast complexes. Of course, I also understand if the authors prefer to show their primary data in the main figures.

We agree that the comparison to the yeast structure would be more important to show in the main text. We have moved the UBA comparison to the Extended Data File (Extended Data Fig. 11) and made the corresponding changes in the main text.

Reviewer #2 (Remarks to the Author):

The manuscript by Sinning and coworkers report on the cryo-EM structures of human NatA bound alone to the 80S ribosome, and together with Methionine Aminopeptidase 2 (MAP2) and MAP1 plus the Nascent Polypeptide Associated Complex (NAC), all in the absence of nascent chain assembly. A comparison of the structures reveals that (1) NatA binds dynamically to a 'distal' binding site on the ribosome that does not interfere with MAP1 or MAP2 binding nor with most other ribosome-associated factors (RAFs), (2) MAP2 binding appears to constrain the conformational freedom of NatA but binds in a position that appears to be incompatible with N-terminal acetylation as MAP2 binds on top of the Protein Tunnel Exit (PTE), (3) NatA binding appears to effect the positioning of the B-arm of the long 28S rRNA expansion ES27L (from the OUT to the IN position), (4) NAC binds at a position between NatA and MAP1, and although not visible in the current map, the NAC α N-terminus points to the Naa15 subunit of NatA and the NAC β C-terminus points towards MAP1 and is thus inferred to bridge communication between NatA and MAP1, (5) NAC α and HYPK employ conserved UBA-like domain to contact distinct sites on the Naa15 subunit of NatA.

We thank reviewer #2 for this summary and assessment of our manuscript. We agree with most of the points, but must politely disagree with point (2), which states that MAP2 binding would be incompatible with N-terminal acetylation by NatA. The central placement of MAP2 on the PTE would allow MAP2 to quickly carry out NME as soon as a nascent chain emerges. As shown in previous structures (Klein et al. 2024) and discussed in our manuscript, MAP2 can undergo a dynamic rotation around its insert domain. This ability of MAP2-like proteins to pivot away from the ribosome has also been described for Ebp1 (Bhaskar et al. 2021). Such a backwards rotation, perhaps promoted or supported by the growing nascent chain, could in our opinion very well explain how access to the nascent chain N-terminus could be granted to NatA to subsequently carry out NTA.

Although the current studies appear to map, and confirm previous studies, on the binding positions of NatA, NAC and MAP1 and 2 to the ribosome how these proteins coordinate NatA activity on the ribosome is not elucidated by these studies. Thus, the conceptual advance of the current study is limited.

We respectfully but strongly disagree with this statement. Our study provides a multitude of novel insights into co-translational protein maturation, goes far beyond the confirmation of previous studies and provides a profound basis for the interplay of the different factors at the ribosome. Before one can dive further into the coordination of these factors, we have to analyze them individually.

To clarify we have summarized our main contributions below:

- 1) The binding position of NatA (namely the 'distal' site) was not previously known. Therefore, our structural insights do not just confirm previous studies, but reveal a new binding site on the ribosome.

- 2) We show that the newly identified NatA binding site differs strongly from the binding site of yeast NatA, and that this change in binding site is in large parts a consequence of the different ribosomal architecture (namely ES7L and ES27L) of yeast and human ribosomes.
- 3) We show that NatA is in a similar unique position as NAC, having a non-intrusive off-center binding site that allows concurrent binding of most other ribosome associated factors. At its 'distal' position NatA has the potential to coordinate with other factors to form multi-protein complexes, similar to NAC. This is important as NatA is responsible for the majority of NTA and needs NME by a MAP for acetylation of its substrates.
- 4) The structures provided in our study show that both methionine aminopeptidases (MAP1 and MAP2) can bind the ribosome together with NatA to form multi-protein assemblies on the ribosome. To date, there are no published structures that showcase a direct coordination between NME and NTA on the ribosome.
- 5) We show that NAC engages in a complex scaffolding function in between NatA and MAP1 and reveal that the C-terminus of NAC would clash with HypK, hinting at a possible competition between the two factors at the ribosome.
- 6) We describe a dynamic motion that NatA, NAC and MAP1 can undergo at the tunnel exit.
- 7) We show that NatA, NAC, MAP1, and MAP2 can all bind the ribosome without a nascent chain substrate. This observation suggests that multi-protein assemblies can compile on the PTE before a nascent chain emerges, in preparation for their enzymatic function.

Other major points are:

1. In addition, given that NatA appears to be dynamically bound to the ribosome with limited resolution and a clear preferred orientation issue with all the structures and particularly with the NatA/NAC/MAP1 structure (panel e of Extended Figure 8), it is unclear to me how the authors derive the detailed protein-protein interaction cartoon and stick models that are shown throughout the manuscript. Such protein-protein interactions need to be supported by showing the corresponding models placed within the cryo-EM densities. Biochemical studies could also support the proposed model for NAC by NatA.

It is true that the resolution of our maps is limited (as in all structures of dynamically bound factors at the ribosome) and that side chain information is lacking. However, the shape of the UBA domain is well defined in our map and the UBA domain could be confidently placed into the signal as a rigid body. We also thoroughly assessed the interaction between the NAC α contact helix and Naa15 α 8-10 and build the most probable model that can biochemically explain the interaction that we see.

To further validate our model, we have performed Alphafold 3 predictions of the NAC-NatA interaction and were delighted to find that the predicted interaction between the NAC α -cH and NAC-UBA with Naa15 agrees with our model (helix register and side chain positions). Both the UBA domain and NAC α -cH are positioned and oriented in the same way. To support our model, we have added the Alphafold prediction and confidence plot to the Extended Data (Extended Data Figure 6).

Given the lacking side chain information, we do however agree with reviewer #2 that these details might be misleading. We followed the recommendation and removed the side chain information from the main text figures. Instead, we only show an indication of side chains by replacing them by alanines in Figure 2d, 4b and Extended Data Figure 11.

In Figure 2a-c, we kept the side chains in these figure panels to show that cationic residues are enriched in areas that face the anionic rRNA backbone, but we do not indicate specific contacts.

In addition to these changes in Figures 2, 4 and Extended Data Figure 11, we have added two more Extended Data Figures (3 and 12) to show the signal of the map superimposed with our models.

2. How was HYPK incorporated into the complex? This is not discussed. Was HYPK included in all to the NatA reconstructions?

We apologize that we did not make this clear. HypK was not added in any of the cryo-EM datasets. For the *in vitro* reconstructions with 80S ribosomes, we added stoichiometric heterodimeric NatA (Naa15+Naa10). Figures that include HypK were generated by superimposing the structure of NatA-HypK (Gottlieb and Marmorstein, 2018), as stated in the figure legends.

3. While the study clearly reveals that NAC and HYPK bind to different surfaces of Naa15, how NAC relieves HYPK-mediated inhibition of NatA is not clear from this study.

While the UBA domains of HypK and NAC α clearly contact Naa15 on a different surface, the contact Helix (cH) of both proteins utilizes the same binding surface in parallel to Naa15 helices α 8-10. Consequently, NAC would clash with HypK on the surface of Naa15, indicating that HypK would need to rearrange to allow the NAC-NatA interaction.

Bound in the distal site, NatA TPRs 1-2 are bend in opposing direction as in the NatA-HypK structure (Gottlieb and Marmorstein, 2018; Weyer et al., 2017). Additionally, it has been shown that HypK strongly stabilizes the NatA complex (Gottlieb and Marmorstein, 2018; Weyer et al., 2017), indicating that NatA becomes more rigid when in complex with HypK. HypK binding to the NatA complex blocks enzymatic activity of Naa10 (Gottlieb and Marmorstein, 2018; Weyer et al., 2017). However, in an *in vivo* context, HypK appears to be essential for proper NatA activity (Arnesen et al. 2010, Gong 2022, Miklánková 2022). From these observations, we draw the conclusion that HypK might inhibit ribosome binding and Naa10 catalyzed acetylation, unless HypK is remodeled in some way. As we identified that the NAC α contact helix (cH) binds in the same position as the HypK-cH, it appears likely that NAC would compete with HypK at the ribosome and might be involved in its remodeling. However, we do not have data with HypK and NatA at the ribosome and therefore only mentioned this possibility as part of the discussion.

4. The distinction that the authors make about a 'distal' NatA binding site is not clear. Is there an alternative 'proximal' site? This needs to be elaborated.

We refer to this newly identified NatA binding site as ‘distal’ site for the following reasons:

- 1) The first and so far only structure of a NatA/E on the ribosome is available from yeast, which positions Naa10 close to the PTE (Knorr et al. 2019). Compared to this, our human NatA binds further away from the PTE, therefore referred to as ‘distal’ site. There, NatA does not interfere with most other ribosome associated factors that bind in closer proximity to the PTE (around universal adapter site 1 and 2 (UAS1&2)). As already outlined above: this is important as NatA is responsible for the majority of NTA and needs NME by a MAP for acetylation of its substrates.
- 2) Human NatE likely has a different binding site similar to yeast NatE. When superimposing the structure of human NatE (Deng et al. 2020) onto our structure of NatA in the distal site, Naa50 would be located far away from the tunnel exit (as shown in Extended Data Figure 8). This position would not make much physiological sense. This indicates that there is likely a ‘proximal’ NatA binding site from which Naa50 can be coordinated.

5. Pg 5, Can the authors rationalize why NatA is better ordered in the MAP2 containing complex than NatA alone if there are no visualized or proposed contacts between the two proteins?

In our dataset of NatA at the ribosome in the absence of other factors, NatA binding is extremely dynamic. The long flexible linkers that connect Naa15 to the α 34 anchor allow NatA to rotate and rappeel towards the PTE. This strong heterogeneity results in a poorly resolved map in our NatA-80S dataset. The presence of MAP2 simply limits the NatA rotation to a smaller angle. With MAP2 present on the ribosome, NatA cannot rotate as far towards the PTE, as it can in the absence of MAP2. As a consequence, the position of NatA is more consistent throughout the individual particle images and allows a higher-resolution reconstruction of NatA in the distal site.

6. It’s not clear from the discussion or figures why the ES27Lout position is incompatible with NatA binding. Can this be better explained? The significance of the ES27L configuration on function is also unclear.

We agree that it is not proven that the ES27Lout position would be incompatible with NatA binding. In our previously reported structure of MAP2 on the human ribosome, ES27L was recruited to the ES27Lout position (Klein et al. 2024). In complex with NatA, MAP2 no longer recruits the expansion segment. As we cannot draw conclusions on the function on ES27L in the context of NatA on the ribosomes, we have decided to move Figure 3 to the supplementary information and made corresponding changes in the main text.

7. If endogenous NAC was already bound in the complex, why did the authors need to reconstitute with recombinant NAC for SPA cryo-EM?

In a preliminary dataset, we indeed found endogenous NAC located at the PTE in complex with NatA and MAP1. However, the population of particles that had NAC on the tunnel exit was highly underrepresented. Our ribosome preparations were

performed at 500 mM KOAc. This high salt concentration likely displaced a large fraction of endogenous NAC. To enrich NAC on the ribosome, to have more control over the sample composition and to have more particles at our disposal for classification, we purified NAC to reconstitute the NatA-NAC-MAP1-80S complex *in vitro*.

Minor points

Pg 2, change “The zinc-finger motive containing” to “The zinc-finger motif containing”

Thank you for catching this typo. We have made the corresponding change in the manuscript.

Extended Data Fig. 3 should be rotated 90 degrees clockwise.

As per request, we have rotated Extended Data Fig. 3

Reviewer #3 (Remarks to the Author):

As the nascent polypeptide chain emerges from the ribosomal tunnel, it is subject to a number of co-translational modifications, including N-terminal excision and acetylation, which are essential in higher eukaryotes. Despite decades of research, the interplay between the different enzymes that mediate these processes at the tunnel exit is not well-understood. The study of Klein et al reports structural insight into the interplay of MAP1, MAP2, NatA and NAC on the human 80S ribosome. The authors show that unlike yeast NatE, human NatA binds in a more distally located position at the tunnel exit that does not interfere with binding of other RAFs, which is illustrated by additional cryo-EM structures of NatA with NAC and MAP1, or MAP2. The manuscript is generally well-written and presented and the results will be of general interest to researchers in the fields of translation and co-translational protein folding/processing. I have two comments that should be addressed and a few minor points.

We thank reviewer #3 for the positive feedback on our manuscript.

Comments.

1. The first comment is that it is a little disappointing that authors use vacant 80S ribosomes for their studies rather than making the effort to form more physiological functional complexes i.e. translating ribosomes with nascent polypeptide chains. After all, the factors under study are all involved in processing/targeting of the nascent polypeptide chain. It seems somewhat contradictory to highlight in the abstract that “...how these enzymatic processes are coordinated in the context of a rapidly translating ribosomes has remained elusive”...and then to talk about multi-enzyme assemblies on 80S ribosomes...as though these are in anyway related to translating ribosomes...At a minimum the authors should add “vacant human 80S ribosomes” and also mention this limitation clearly in their manuscript discussion.

We see this point of the referee, however, we do not agree that interaction studies on idle 80S ribosomes are not related to translating ribosomes. It is clear from previous and current studies that idle 80S ribosomes are well suited to study the interaction with ribosome associated factors. Therefore, we have outlined our reasoning for the use of non-translating 80S ribosomes:

- 1) Translation progresses rapidly and the N-terminal residue is processed almost immediately after it emerges from the ribosome. The enzymes that are involved in this process (e.g. MAP1, MAP2, NatA) likely assume their position at the PTE before a nascent chain is presented. This is important as these factors are underrepresented compared to ribosomes. In our opinion, it is essential to determine the structure of this preparative 'start assembly'! We certainly agree that structures with a nascent chain can be useful. However, without the structure on vacant ribosomes, there would be nothing to compare the RNC structure with. Conclusions on how the nascent chain influences the ribosome associated factors would not be possible without the 'vacant' reference structures. Examples where such comparisons are possible are growing and support our reasoning (see Point 4).
- 2) In addition, there are hundreds of nascent chain substrates that qualify for NME and NTA. Each nascent substrate will have slightly different biophysical properties and might influence the ribosome associated complexes in different ways. We believe that choosing one particular substrate at random and determining the structure should not be seen as the 'baseline' for this particular assembly. Instead, we find it more useful to first define the structure of NAC, MAP1, MAP2 and NatA containing complexes without a nascent chain. Further studies can then build on these structures and introduce different nascent substrates to investigate how they might influence the 'start assemblies'.
- 3) Furthermore, one should be aware that the nascent chain was not visible beyond the tunnel exit in most RNC structures with ribosome associated complexes. Examples include the structure of yeast NatA on the ribosome (Knorr, 2019), NAC on the ribosome (Jomaa, 2022), NAC-MAP1 on the ribosome (Gamerding, 2023) as well as structures with RAC (Cheng et al. 2022).
- 4) In addition, in those cases where a comparison is possible between idle ribosomes and RNCs, the nascent chain had no influence on the positioning of the ribosome associated factor at the ribosome. Examples related to our structures include NAC (Jomaa, 2022) and the complex of NAC and MAP1 (Gamerding, 2023).
- 5) Finally, we would not per se consider programmed, stalled ribosomes as particularly physiological samples. The process of obtaining such samples is quite stressful for a cell and stalling triggers a complex cascade of processes involving the ribosomal quality control machinery. For RNCs derived from *in vitro* translation one could argue that choosing the identity and length of the mRNA and the amount/ratios of specific factors in the assay does not reflect the complex *in vivo* situation, but provides a single snapshot. Therefore, these approaches are important tools, but also have their limitations.

Nonetheless, we agree with reviewer #3 that structures of RNCs in complex with NAC, MAP1, MAP2 and NatA might offer additional valuable information about the

coordination of NME and NTA. To address this, we have added two sentences to the discussion related to the use of vacant ribosomes.

2. The second comment is about the absence of non-segmented cryo-EM density images for the complexes in Figure 1. These should be added to this figure with the models underneath. The other point is that the average and local resolutions of the different factors in the complexes should be clearly states in the main text, rather than hidden away in the ED Figure file. There are local resolution images in the ED however the authors need to add transparent density images with molecular models next to them so the reader can understand which regions of the maps have which resolution. The blobby looking density in Figure 3 does not instill confidence, but presumably these are filtered maps – but nothing is stated in the legend. Perhaps the authors should provide some density images to convince the reader that the sidechain interactions they describe, for example, in Figure 2, 4 and 5, are supported by their density and local resolution.

We agree that figures which include the signal of the maps would help to interpret our data. We have tried to incorporate this in Figure 1 in the main text but we find that the figure becomes much less readable. As an alternative, we have created two large Extended Data Figures (3 and 12) that showcase the signal of the map and highlights some areas of interest, namely the NAC α -UBA domain and NAC α -cH.

We also agree that it would be helpful to reference the local resolution within the main text and have incorporated this information in the manuscript.

Some minor points:

1. Line 43 and 324, why is the initiator methionine in a highly predictable position. I could imagine it is highly flexible and the location is not able to be predicted at all.

When translation has just begun, the nascent chain is conformationally confined within the tunnel exit. When the nascent chain grows to a length of around 40 amino acids the N-terminal methionine is in a predictable position because it will emerge first from the exit tunnel. For longer nascent chains and with the onset of protein folding, the position of the N-terminus will be different for different proteins (and no longer “predictable”) and it would become more challenging to recognize and modify this residue.

2. Line 77, “as a the”

Thank you for catching this typo. We have made the corresponding change in the manuscript.

3. The colors for Naa15 and rRNA seem very similar and hard to distinguish. The authors should change this...perhaps grey as they used in their model in figure 6?

We agree that a different color for the rRNA would make the figures easier to read. We followed this suggestion and changed the color to grey.

4. Line 241, "Alike"

This is not a typo (Alike, meaning similar to, or equal to)

REVIEWER COMMENTS

Reviewer #2 (Remarks to the Author):

The authors have adequately addressed my concerns.

The authors should address the relatively minor issues below:

Line 115, change to “active centers are different”

The comparison between the human and yeast structures in Figure 3 is not possible to see as currently depicted. Can the authors adjust the colors between the human and yeast and indicate on the Figure or legend which is which?

Reviewer #3 (Remarks to the Author):

We see this point of the referee, however, we do not agree that interaction studies on idle 80S ribosomes are not related to translating ribosomes. It is clear from previous and current studies that idle 80S ribosomes are well suited to study the interaction with ribosome associated factors. Therefore, we have outlined our reasoning for the use of non-translating 80S ribosomes:

I don't really like the use of the term "idle" since it has an implication that the ribosome was translating and has paused, i.e. like a car which is idling at a intersection before starting to drive again. I understand the reasoning of the authors, but am still disappointed to look at all the factors in complex with vacant ribosomes rather than actually translating ones i.e. the physiological situation that would be in the cell.

We agree that figures which include the signal of the maps would help to interpret our data. We have tried to incorporate this in Figure 1 in the main text but we find that the figure becomes much less readable. As an alternative, we have created two large Extended Data Figures (3 and 12) that showcase the signal of the map and highlights some areas of interest, namely the NAC²-UBA domain and NAC²-cH. We also agree that it would be helpful to reference the local resolution within the main text and have incorporated this information in the manuscript.

I find it still inappropriate to have a structural paper that does not have a single image with the cryo-EM map density. Particular in this case where the resolution is not particularly impressive...most of the factors are in the 6-8Å range and it would be appropriate for the reader to see this immediately in the main images so that the reader can understand the extent to which the authors can interpret their structures.

As I wrote previously the authors should also include additional images next to their local resolution images that have transparent density with the model inside so the reader can actually understand what the local resolution maps are pertaining to. Moreover, the quality of the maps appears not to be sufficient to describe any sidechain information, yet many figures show sidechains. Given the poor local resolution, the authors need to provide density images for these sidechains to convince the reader that such information is present in their maps.

“Alike” might be a word but this is not the way to use it. One might say something like “These two things are alike” which is equivalent to “these two things are similar. But one would not write “Similar NatA, NAC...”.

REVIEWER COMMENTS

Reviewer #2 (Remarks to the Author):

The authors have adequately addressed my concerns. The authors should address the relatively minor issues below:

Line 115, change to “active centers are different”

Thank you for catching this typo. We have made the corresponding change in the manuscript.

The comparison between the human and yeast structures in Figure 3 is not possible to see as currently depicted. Can the authors adjust the colors between the human and yeast and indicate on the Figure or legend which is which?

We agree that this would be helpful. We have changed the figure according to your recommendation.

Reviewer #3 (Remarks to the Author):

We see this point of the referee, however, we do not agree that interaction studies on idle 80S ribosomes are not related to translating ribosomes. It is clear from previous and current studies that idle 80S ribosomes are well suited to study the interaction with ribosome associated factors. Therefore, we have outlined our reasoning for the use of non-translating 80S ribosomes:

I don't really like the use of the term "idle" since it has an implication that the ribosome was translating and has paused, i.e. like a car which is idling at a intersection before starting to drive again. I understand the reasoning of the authors, but am still disappointed to look at all the factors in complex with vacant ribosomes rather than actually translating ones i.e. the physiological situation that would be in the cell.

We addressed this concern and changed the word 'idle' to 'vacant' throughout the manuscript.

We agree that figures which include the signal of the maps would help to interpret our data. We have tried to incorporate this in Figure 1 in the main text but we find that the figure becomes much less readable. As an alternative, we have created two large Extended Data Figures (3 and 12) that showcase the signal of the map and highlights some areas of interest, namely the NAC α -UBA domain and NAC α -cH. We also agree that it would be helpful to reference the local resolution within the main text and have incorporated this information in the manuscript.

I find it still inappropriate to have a structural paper that does not have a single image with the cryo-EM map density. Particular in this case where the resolution is not particularly impressive...most of the factors are in the 6-8Å range and it would be appropriate for the reader to see this immediately in the main images so that the reader can understand the extent to which the authors can interpret their structures.

As requested, we have overlaid the cryo-EM map in all panels of Figure 1, as well as the central panel of Figure 2. In addition, we have kept Extended Data Figures 2, 3, 10 and 12 to communicate the quality of the map, as well as numerous in-text references to the resolution. Combined, we feel that these efforts should make it easy for the reader to interpret our data in context of the resolution.

As I wrote previously the authors should also include additional images next to their local resolution images that have transparent density with the model inside so the reader can actually understand what the local resolution maps are pertaining to.

We have added images of the models with transparent cryo-EM maps next to the local resolution images for better orientation.

Moreover, the quality of the maps appears not to be sufficient to describe any sidechain information, yet many figures show sidechains. Given the poor local resolution, the authors need to provide density images for these sidechains to convince the reader that such information is present in their maps.

After the first round of revision, we had removed detailed side chain information from Figure 2d, Figure 4 and Extended Data Figure 11, and instead only show representative alanines to indicate interactions. Because of the inherent dynamic interaction of NatA, NAC and MAP1 at the ribosome, side chains are not resolved. However, the overall position and orientation of these factors is well defined in the map and many interaction surfaces can be modelled between the factors and the ribosome.

Nonetheless, we do not want our figures to be misleading in any way and have now also removed the representative alanines from Figures 2d, 4 and Extended Data Figure 11.

The only side chains that are left in any figures are shown in Figure 2a-c. Here, we show that arginines and lysines are enriched on the rRNA facing side of Naa15-TPR1 and Naa15- α 34. While we do not see each sidechain defined in the map, the orientation of the Naa15 helices relative to the ribosome is absolutely clear from our data. A different orientation would not make sense and therefore we are confident that the representation of these residues is appropriate and important. Since we do not have 'side-chain resolution', we do not label specific residues or indicate that any particular residue is more or less important than another. We merely show exemplary residues that are clearly oriented towards the ribosomal surface.

To address this concern, we have added a sentence to the results to mention that individual side chains are not resolved in the map.

"Alike" might be a word but this is not the way to use it. One might say something like "These two things are alike" which is equivalent to "these two things are similar. But one would not write "Similar NatA, NAC...".